# Candidalysin Is the Hemolytic Factor of *Candida albicans*

**DOI:** 10.3390/toxins14120874

**Published:** 2022-12-15

**Authors:** Selene Mogavero, Sarah Höfs, Alexa N. Lauer, Rita Müller, Sascha Brunke, Stefanie Allert, Franziska Gerwien, Sabrina Groth, Edward Dolk, Duncan Wilson, Thomas Gutsmann, Bernhard Hube

**Affiliations:** 1Department of Microbial Pathogenicity Mechanisms, Leibniz Institute for Natural Product Research and Infection Biology–Hans Knoell Institute (HKI), 07745 Jena, Germany; 2Institute of Microbiology, Friedrich Schiller University, 07745 Jena, Germany; 3Research Center Borstel, Division of Biophysics, 23845 Borstel, Germany; 4QVQ B.V., 3584 CL Utrecht, The Netherlands; 5Medical Research Council Centre for Medical Mycology, University of Exeter, Exeter EX4 4RN, UK

**Keywords:** hemolysin, *Candida albicans*, candidalysin, erythrocytes, peptide toxin, intercalation, PPADS, neutralization, nanobodies, VHH

## Abstract

*Candida albicans* produces an important virulence factor, the hypha-associated *Ece1*-derived secreted peptide toxin candidalysin, which is crucial for the establishment of mucosal and systemic infections. *C. albicans* has also long been known to be hemolytic, yet the hemolytic factor has not been clearly identified. Here, we show that candidalysin is the hemolytic factor of *C. albicans*. Its hemolytic activity is modulated by fragments of another *Ece1* peptide, P7. Hemolysis by candidalysin can be neutralized by the purinergic receptor antagonist pyridoxal-phosphate-6-azophenyl-2′,4′-disulfonic acid (PPADS). PPADS also affects candidalysin’s ability to intercalate into synthetic membranes. We also describe the neutralization potential of two anti-candidalysin nanobodies, which are promising candidates for future anti-*Candida* therapy. This work provides evidence that the historically proposed hemolytic factor of *C. albicans* is in fact candidalysin and sheds more light on the complex roles of this toxin in *C. albicans* biology and pathogenicity.

## 1. Introduction

The opportunistic fungal pathogen *Candida albicans* is a common colonizer of human epithelia, with its major reservoir being the gut [1]. Predisposing conditions allow this fungus to invade the intestinal barrier and reach deeper tissues via dissemination within the bloodstream [2,3,4]. A key factor that mediates *C. albicans* translocation through the intestinal epithelium is its secreted peptide toxin candidalysin [4], the first peptide toxin discovered in a human fungal pathogen and critical for mucosal and systemic infections [4,5,6,7,8,9].

Candidalysin is generated from the longer polyprotein Ece1, which is processed by the Golgi-located Kex2 protease into eight peptides with C-terminal lysine–arginine (KR) residues [5,10]. It was subsequently shown that a second processing event occurs by another Golgi-located protease, Kex1 [5,11] which produces mature candidalysin by removing the C-terminal arginine from P3 [5,11]. Peptides deriving from a Kex2-only processing are secreted, albeit at lower levels than mature candidalysin [5,11]. Both P3 (ending with KR) and candidalysin (ending with K) have cytolytic activity against epithelial cells [5].

In 1993, Birse et al. identified, cloned and characterized the *ECE1* gene in *C. albicans* [12]. In the following decades, *ECE1* was known to be one of the most highly transcribed genes during hypha formation of *C. albicans* and frequently used as a marker gene for hypha formation [12,13,14].

Although hyphae were already well recognized as the invasive morphology, and hypha-associated genes predicted to contribute towards virulence, it was a further 23 years before the biological function of Ece1 was discovered [5].

Following the original identification of *ECE1*, an independent study by Manns et al. (1994), showed that *C. albicans* produces a hemolytic factor [15]. In 1997, it was shown that hemoglobin is utilized by *C. albicans* hyphae but not yeast [16], and a few years later, Watanabe et al. (1999) described characteristics of a hemolytic factor in the culture supernatant of *C. albicans* [17]. Several further studies reported that clinical isolates of *C. albicans* showed hemolytic activities [18,19,20]. However, the mechanistic basis of *C. albicans* hemolytic activity has remained unknown.

Given its pleiotropic effects on several cell types, we hypothesized that candidalysin is the unknown hemolytic factor and investigated its activity of candidalysin toward erythrocytes in more detail. In the current study we demonstrate that candidalysin is the hitherto unidentified hemolysin of *C. albicans* and show that its hemolytic activity is enhanced by divalent cations and blocked by purinergic receptor inhibition.

## 2. Results

### 2.1. Ece1 Is Responsible for Human RBC Lysis by C. albicans Hyphae

To investigate the impact of *C. albicans* and the hyphal-associated protein *Ece1* on hemolysis, we incubated different *ece1* mutant strains (Table 1) in the presence of washed red blood cells (RBCs) for 24 h.

The alkaline pH of the cell culture medium (RPMI) in which RBCs were resuspended and temperature of 37 °C induced *C. albicans* hypha formation during co-incubation with the RBCs (Figure 1a). The isogenic wild-type *C. albicans* strain had a strong hemolytic effect (Figure 1b). Both *ece1* mutant strains, lacking either the whole *ECE1* open reading frame, or just the P3 coding sequence, were unable to lyse RBCs above the background levels observed in the vehicle control (medium only). The revertant strain, containing one of the two copies of the *ECE1* gene at a heterologous locus, was able to rescue the wild-type phenotype, up to approximately 50% of wild-type levels.

Therefore, *ECE1* and the P3 coding sequence are essential for *C. albicans*-mediated hemolysis.

### 2.2. Candidalysin Is Also a Hemolytic Peptide Toxin

During the processing of Ece1, the precursor of candidalysin, P3, and all other Ece1 peptides are sequentially processed by Kex2 followed by Kex1 digestion. The reason for this additional Kex1 processing is not clear. To investigate the hemolytic potential of all peptides, including P3, and candidalysin, we first incubated the products of Kex2-digested Ece1 [10], P2 to P8 (P1 was excluded since it is predicted to be functional to the signal peptide, SP) with RBCs (Figure 2a). As predicted, of the Ece1 peptides, only P3 caused hemolysis (Figure 2b).

Next, we directly compared the lytic activity of P3 to the fully mature candidalysin. Figure 2c,d show that increasing concentrations of either peptide (range 1–8 µM) correlated with increasing lysis of RBCs and, at the lower concentrations, a time-dependent hemolysis can be observed. Candidalysin is clearly less hemolytic than P3 at all tested concentrations (Figure 2c vs. Figure 2d). At concentrations of 2 µM and above, almost full RBC lysis was observed at the earliest time point of 5 min incubation, for P3. The same happened with candidalysin, but at the concentration of 4 µM and above.

### 2.3. The Hemolytic Potential of Candidalysin and P3 Is Ion-Dependent and pH-Independent

Many hemolytic toxins exhibit increased activity in the presence of ions [22,23]. We therefore tested if chelating divalent cations would influence the hemolytic potential of candidalysin and/or P3. In the presence of 100 mM ethylene-diamine-tetra-acetic-acid (EDTA) the hemolytic potential was drastically reduced for both peptides, more than 4.4-fold reduction for P3, and a 4.2-fold reduction for candidalysin. Adding back magnesium ions in the form of MgCl_2_ rescued the hemolytic potential, up to the point of exceeding the control hemolysis values when adding 100 mM MgCl_2_ (Figure 3a). In contrast, hemolysis was not affected by alterations in pH (pH 5 to 8), with a significant difference only observed for candidalysin between pH 5 and pH 7 (Figure 3b).

### 2.4. Ece1 P7 Derivatives Modulate the Hemolytic Potential of Candidalysin

Along with candidalysin, several Ece1 peptides are secreted when *C. albicans* is cultured in hyphae-inducing conditions [5]. Candidalysin is the most abundant of these, followed by fragments of peptide 7, which we have termed P7a and P7b (Figure 4a) [5,11]. Our current working hypothesis is that non-candidalysin Ece1 peptides (NCEPs) modulate the activity of candidalysin. Since we know that Kex1 processing removes the C-terminal arginine, a likely intermediate precursor of P7a and P7b is P7K (same as P7 but without the last R). We, therefore, tested the influence of these peptides on the hemolytic potential of candidalysin. The addition of P7K resulted in a 31% increase in the hemolytic activity of candidalysin (*p* = 0.0143); P7a also increased candidalysin activity—by 16%; however, this was not statistically significant (*p* = 0.0541). Interestingly, whilst P7K alone had no effect, the P7a fragment did exhibit low-level cytolytic activity (Figure 4b).

### 2.5. Purinergic Receptor Antagonists Protect RBCs from Candidalysin-Induced Lysis

Hemolysins such as the α-hemolysin of *Escherichia coli*, or the α-toxin of *Staphylococcus aureus*, induce RBC lysis by activation of a purinergic signaling cascade from P2X receptors [24,25]. Using the purinergic receptor P2X antagonist pyridoxal-phosphate-6-azophenyl-2′,4′-disulfonic acid (PPADS) we found that candidalysin-induced hemolysis is reduced in the presence of this inhibitor (Figure 5a). To verify if PPADS acts exclusively via purinergic signaling, or if there is an unspecific inhibitory effect, we tested this compound with candidalysin and artificial membranes. To our surprise, PPADS was able to inhibit permeabilization of artificial membranes (containing no receptors) by candidalysin. This observation indicates that PPADS can have a non-specific effect on hemolytic activity which is independent from its well-characterized role in purinergic signaling inhibition (Figure 5b).

### 2.6. Llama-Derived Anti-Candidalysin Nanobodies Neutralize the Hemolytic Potential of Candidalysin

In our recent study, we developed two nanobodies (CAL1-F1 and CAL1-H1) which bind to both synthetic and *C. albicans*-secreted candidalysin [26]. Here, we tested the neutralizing potential of such nanobodies. Two independent clones were pre-incubated with candidalysin for 1 h, to allow the nanobodies to bind the toxin. After a further 1 h exposure to RBCs, hemolysis was measured as described above. Both clones neutralized hemolysis by candidalysin, but clone CAL1-H1 was much more efficient (Figure 6a): complete inhibition of hemolysis was reached when candidalysin and CAL1-H1 were incubated at a 1:1 ratio. For CAL1-F1, when the nanobody was added at a 2:1 ratio with candidalysin, hemolysis was reduced by approximately 50%. The nanobodies were also tested for toxicity vs. RBCs, and for activity towards melittin, a peptide toxin with similar properties to candidalysin. Neither nanobody caused hemolysis on their own, and when incubated with the bee venom component melittin, there was some degree of neutralization of damaging potential (17% for CAL1-F1 and 30% for CAL1-H1) at the highest concentration of nanobody used in this experiment, i.e., 8 µM (Figure 6b).

## 3. Discussion

The first observations that *C. albicans* exhibited hemolytic activity date back to 1951 [27], and since then several studies have confirmed this activity [15,16,17,28], but none were able to identify the hemolytic factor of *C. albicans*.

Watanabe et al. characterized the hemolytic factor as a sugar moiety of a *C. albicans* mannoprotein, but the exact identity of this protein and the molecular mechanism underlying RBC lysis were not determined [17].

In a previous attempt to demonstrate candidalysin-dependent hemolysis, we incubated wild type *C. albicans* in parallel with candidalysin-defective mutants on blood agar plates, prepared following the recipe by Manns et al. [15]. We were not able to observe any difference in the hemolysis produced by the strains, which is described as a lightened (yellow) and transparent halo around the *Candida* colonies. Both the wild-type and the mutant strains did produce beta-hemolysis (data not shown). We argue that the hemolytic halo observed around both strains is caused by other factors, possibly secreted aspartic proteases (Saps), that easily manage to lyse the already fragile RBCs present in blood agar. As a matter of fact, in order to prepare blood agar, RBCs need to be added to the still liquid agar while at temperatures above 45 °C [29]. However, evidence suggests that at those temperatures the RBC membrane’s stability is reduced [30].

To circumvent the limitation of using blood agar to assess hemolysis, and to obtain a more quantifiable read-out, we set up an in-liquid hemolysis assay. The experimental conditions adopted allowed the fungus to filament (Figure 1a), a necessary condition to study the contribution of candidalysin [5]. Hemolysis was completely absent with the candidalysin-lacking mutants (*ece1*Δ/Δ and ΔP3), providing definitive proof that candidalysin is the hemolytic factor (Figure 1b). This conclusion is further supported by the fact that, similar to what was already observed for oral epithelial cells [5], synthetic candidalysin and its direct precursor P3 are strong hemolytic peptides, whereas the other NCEPs are not (Figure 2a,b).

The involvement of *Ece1* in the lysis of RBCs is further supported by the finding that *ECE1* expression is induced in the presence of hemoglobin [31].

We showed that synthetic candidalysin and its precursor P3 exert a potent cytolytic effect on human RBCs. Using different peptide concentrations and performing a time course experiment, it was demonstrated that both P3 and candidalysin rapidly lyse RBCs, with P3 being active at lower concentrations than candidalysin (Figure 2c,d). This feature might be of interest in the search for bioactive compounds with various applications (e.g., to target tumor cells [32]). The reason for these differences in hemolytic potential and the biological relevance for the removal of the C-terminal arginine (R) remain unknown. We postulate that Kex1-removal of the terminal arginine residue and resultant dampening of cytolytic potential may be an adaptation to avoid excessive host damage which could hyperactivate a detrimental immune response leading to rapid clearing of the fungus.

Concentrated erythrocytes and micromolar concentrations of synthetic peptides were used in the present study. It is highly unlikely that these concentrations are reached in the bloodstream during disseminated candidemia. However, *C. albicans* hyphae have been shown to bind to and rosette erythrocytes [33]. Such proximity may facilitate candidalysin-mediated hemolysis during candidemia. 

From a longer-term evolutionary perspective, *C. albicans* has been regularly exposed to erythrocytes at mucosal surfaces during mucosal bleeding and in the vagina during menstrual discharge. As erythrocytes are a rich source of heme and iron, the fungus may utilize the hemolytic mechanism described in this study to acquire this essential trace mineral. Indeed, *C. albicans* encodes a sophisticated pathway for the extraction of heme and heme-iron from hemoglobin [34].

Candidalysin’s hemolytic activity could also be responsible of producing hemocidins in the vaginal niche through the lysis of RBCs present in the menstrual discharge that would release hemoglobin. Hemocidins are a group of microbicidal peptides arising from heme-binding proteins such as hemoglobin [35]. In vivo, such peptides can be found, among others, in the menstrual discharge [36]. Here, these peptides might strongly influence the composition of the vaginal microbiota and therefore contribute to a *C. albicans* overgrowth, eventually resulting in vaginal candidiasis.

Bocheńska et al. demonstrated that *C. albicans* Saps are able to liberate bactericidal hemocidins from hemoglobin, with the released peptides showing a strong killing activity against *Lactobacillus acidophilus* and, to a lower degree, against *E. coli* [37]. In this scenario, it is plausible that candidalysin plays a role in liberating hemoglobin from RBCs, thus making it accessible to Saps that in turn liberate hemocidins.

Candidalysin/P3-induced hemolysis was shown to be significantly reduced in the presence of the potent chelating agent EDTA. This chelator is known to form stable complexes with divalent cations, e.g., calcium, magnesium, or zinc ions. Divalent cations play an important role in various processes connected to microbial toxins. Heavy metal ions, for example, represent essential cofactors for the activity of metalloproteases, which often act as receptors for pore-forming toxins [38]. Ion-chelation could therefore impair receptor activity, resulting in a reduced toxin function. A further explanation would be the initiation of cellular signaling pathways within the host cell in response to candidalysin/P3. Divalent cations are potent second messengers involved in the activation of various cellular processes [39]. We showed that the addition of excess Mg_2_^+^ ions to cells pre-treated with EDTA restored the cytotoxicity of candidalysin and P3 (Figure 3a). The dependency of cytotoxic function on divalent cations has also been demonstrated for bacterial toxins, e.g., the *Bacillus thuringiensis* Cry1Aa toxin [38].

As *C. albicans* encounters a variety of environmental niches when colonizing or infecting its human host, an effective virulence factor needs to be active in a broad range of different pH values. In isotonic media, a decrease of the pH from 9.4 to 4.8 has been shown to result in an increased RBC volume, a decrease in density and a more spherical shape of these blood cells [40]. Therefore, the hemolytic properties of candidalysin and P3 were tested at a pH between 5 and 8, to prevent the occurrence of pH-induced impairment of RBC viability. Only a minimal difference in lysis efficiency could be observed at the tested pH values (Figure 3b). This is not surprising, as many toxins have been shown to be active across a wide range of pH values. The VacA toxin released by *Helicobacter pylori*, for example, is activated by a short exposure to acidic solutions (pH 1.5–5.5) and once activated remains stable at pH 1.5 and can even resist pepsin digestion at pH 2 at 37 °C [41]. Hypothesizing a role of candidalysin in the generation of menstrual hemocidins, as discussed earlier, full activity may very well be reached at a more acidic pH values, such as pH 4, corresponding to the vaginal pH and to the finding that Sap-mediated hydrolysis of hemoglobin, resulting in the production of hemocidins, was also most effective at pH 4 [37].

The role of the other Ece1 peptides in *C. albicans* virulence is unknown. Here, we show that derivatives of P7 are able to augment candidalysin’s hemolytic activity (Figure 4). One of which, P7a, is, after candidalysin, one of the most abundant peptides found in *C. albicans* hyphal supernatants [5,11].

The presence of P7K and P7a may increase candidalysin aggregation, which may enhance its hemolytic activity. Nevertheless, the impact of P7-derivates was relatively minor and our current study suggests that candidalysin alone remains the major cytolytic factor against all mammalian cell types tested to date, with P7-derivatives possibly modulating its activity.

Previous studies have shown that purinergic receptors are activated by the hemolytic toxins, and subsequent signaling cascades are responsible for the ultimate hemolysis [24,25]. The PPADS purinergic receptor antagonist did indeed reduce candidalysin-induced hemolysis (Figure 5a). However, a more in-depth analysis showed that PPADS can also inhibit the intercalation of candidalysin into synthetic membranes, that do not contain purinergic receptors (Figure 5b). Others have also questioned the theory that purinergic signaling is involved in (amplifying) toxin-induced hemolysis [42]. The mechanism of PPADS-mediated protection of membranes remains unclear. Of note, the more pronounced effect observed in the case of artificial membranes can be explained by the fact that the effective membrane surface area in the case of erythrocytes is significantly higher than that of the planar single membrane layer in the case of a reconstituted membrane.

Our approach to block candidalysin-induced hemolysis with llama-derived anti-candidalysin nanobodies [26] was successful, with both tested nanobody clones able to neutralize the toxin (Figure 6). Clone CAL1-H1, when added in equimolar ratio to candidalysin, was able to completely block hemolysis by candidalysin. Clone CAL1-F1 was not as efficient, although still able to reduce hemolysis. These data suggest that one molecule of CAL1-H1 can neutralize one molecule of candidalysin, whilst even 2:1 equivalents of CAL1-F1 only reduced hemolysis by 47%. This is possibly why CAL1-F1 was more effective for immunofluorescence stainings, as described in our previous paper [26]: more CAL1-F1 molecules can bind a single candidalysin, thereby amplifying the staining signal. Our data indicate that these nanobodies are promising therapeutic agents. *C. albicans* pathogenicity, especially in the vaginal environment, is strictly linked to immunopathology as a consequence of direct exposure to candidalysin [8]. Having a treatment option that only neutralizes the main virulence factor, rather than trying to eradicate the fungus, is ideal as this strategy would not select for resistance.

## 4. Materials and Methods

### 4.1. Fungal Strains and Culture Conditions

*C. albicans* strains were cultured overnight in YPD broth (yeast extract 1% *w/v*, peptone 1% *w/v*, dextrose 2% *w/v*), at 30 °C, shaking at 180 rpm, washed twice with phosphate buffered saline (PBS) and adjusted to the assay-required cell concentration in the respective assay media. Strains used throughout the study are listed in Table 1.

### 4.2. RBC Preparation

RBCs were purified from whole blood of healthy volunteers. Blood was collected in tubes containing EDTA-K3 as an anticoagulant (approximately 9 mL/tube). To separate RBCs from white blood cells (WBCs) and plasma, three washes were performed by centrifugation (2000× *g* for 10 min) and subsequent re-suspension to reach 9 mL. First, the RBCs were resuspended in Dulbecco′s Phosphate Buffered Saline (DPBS) + 2% heat inactivated fetal bovine serum, then in pure DPBS, and eventually in RPMI (for assays with live candida) or again DPBS (for assays with peptides). After each centrifugation step, care was taken to remove as many WBCs as possible from the central buffy layer. A hemocytometer (Auto Hematology Analyzer BC-5300 Vet (Mindray, Shenzhen, China)) was used to verify that the WBC count was negligible (<10^9^/mL) and to determine the RBC concentration. This was then adjusted to 5 × 10^8^ cells/mL, in the respective media or buffer.

### 4.3. Handling of Peptides

Peptides were purchased from two different suppliers, namely Proteogenix (Strasbourg, France) and Peptide Protein Research Ltd. (Southampton, UK). Stock solutions were prepared in pure water at a concentration of 1.4 mM, then aliquoted and stored at −20 °C. Since some peptides, including candidalysin, are hydrophobic and do not go well in solution, the stock solutions were mixed well by vortexing before aliquoting. Working solutions were then prepared in the relative buffer at the final concentration indicated for each assay.

### 4.4. Hemolysis Assay

To assess the hemolytic potential of *C. albicans* strains, RBCs and fungal cells were mixed in a 1:1 ratio (5 × 10^7^ each, in a final volume of 600 µL) in a 2 mL plastic test tube (screw cap). Every condition was performed in duplicates (technical replicates) and with at least three different donors (biological replicates). Test tubes were then incubated horizontally at 50 rpms, 37 °C, for 24 h. To assess the hemolytic potential of peptides, 10^7^ RBCs were incubated with the relevant peptide at the indicated final concentration in a total volume of 150 µL, for 1 h at 37 °C, in a 96-well PCR plate. After incubation, samples were centrifuged (2000× *g* for 3 min) and 100 µL of supernatant was transferred to a fresh 96-well plate to measure hemoglobin absorbance at 414 nm. Hemolysis was defined as the absorbance of the sample relative to the absorbance of the full lysis control sample (RBCs incubated with pure water), following subtraction of the absorbance of the vehicle control (RBCs incubated with media or buffer only).

### 4.5. Determination of Membrane Permeabilization

Tethered bilayer lipid membranes with 10% tethering lipids and 90% spacer lipids (T10 slides) were formed using the solvent exchange technique according to the manufacturer’s instructions (SDx Tethered Membranes Pty Ltd., Sydney, Australia). Briefly, 8 µL of 3 mg/mL DOPC solution in ethanol was added into each of the six channels, incubated for 2 min and then 92 µL buffer (100 mM KCl, 5 mM HEPES, pH 7.4) was added. After rinsing 3× with 100 µL buffer, the conductance and capacitance of the membranes were measured for 20 min before injection of different concentrations of PPADS to the final concentrations in the individual channels as shown. Candidalysin was subsequently added at different concentrations to all six channels (t = 50 s: 5 µM; t = 80 s: 10 µM; t = 110 s: 20 µM). All experiments were performed at 37 °C. Signals were measured using the tethaPod (SDx Tethered Membranes Pty Ltd., Sydney, Australia).

### 4.6. Statistical Analysis

All diagrams show mean values with standard deviations, unless stated otherwise. All tests were performed in biological (e.g., different donors) replicates, each with technical replicates (see figure legends for n). A paired *t*-test was used if the data points stemmed from the same donor. For significance analysis of quantitative data, a two-tailed Student’s *t*-test was generally employed. For multiple comparisons, an ANOVA followed by a Tukey’s test was used to determine statistical significance among the selected experimental conditions. In cases where normality of the data was not self-evident, a Shapiro–Wilk test was performed before the *t*-test. All ratios were log-transformed prior to statistical analysis for their differences (with a minimum value of 0.01). All diagrams give the significance level that was passed as *, *p* < 0.05; **, *p* < 0.01; ***, *p* < 0.005; ****, *p* < 0.001; and ns, not significant.

## Figures and Tables

**Figure 1 toxins-14-00874-f001:**
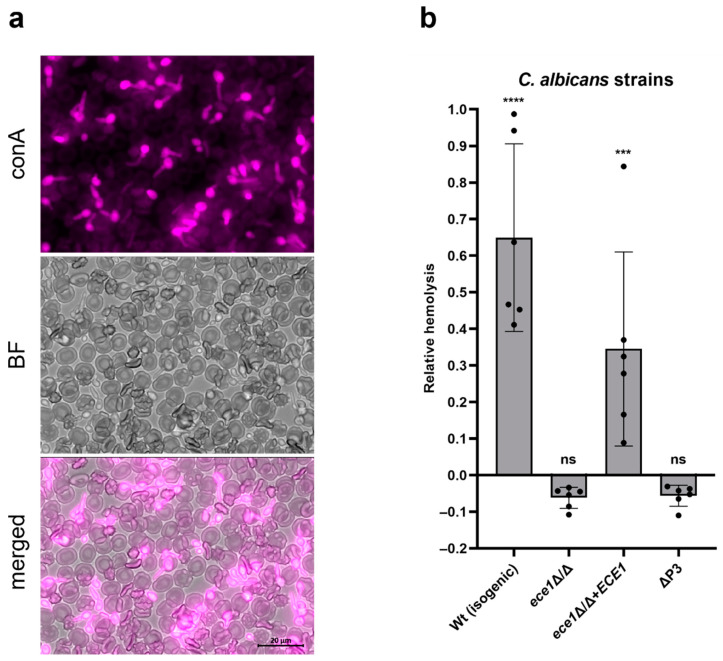
*Candida albicans* lyses red blood cells (RBCs) by means of candidalysin. (**a**) *C. albicans* isogenic wild-type filamenting after 4 h incubation with RBCs. Fungal cells were stained with concanavalin A (conA, magenta). BF, bright field. (**b**) Purified RBCs were incubated with candidalysin-competent or -incompetent *C. albicans* strains for 24 h at 37 °C. Hemolysis was quantified by measuring the absorbance of sample’s supernatant at 414 nm, and plotted relative to the full lysis control sample (RBCs incubated with pure water), following subtraction of the vehicle control. Each data point in (**b**) represents a different donor (average of 2 technical replicates). Error bars show the standard deviation. For statistical analysis, an arbitrary value of 0.01 was assigned to any value that was below this threshold. Student’s paired *t*-tests, vs. the vehicle control, were then performed on log-transformed data. ***, *p* < 0.005; ****, *p* < 0.001; ns, not significant.

**Figure 2 toxins-14-00874-f002:**
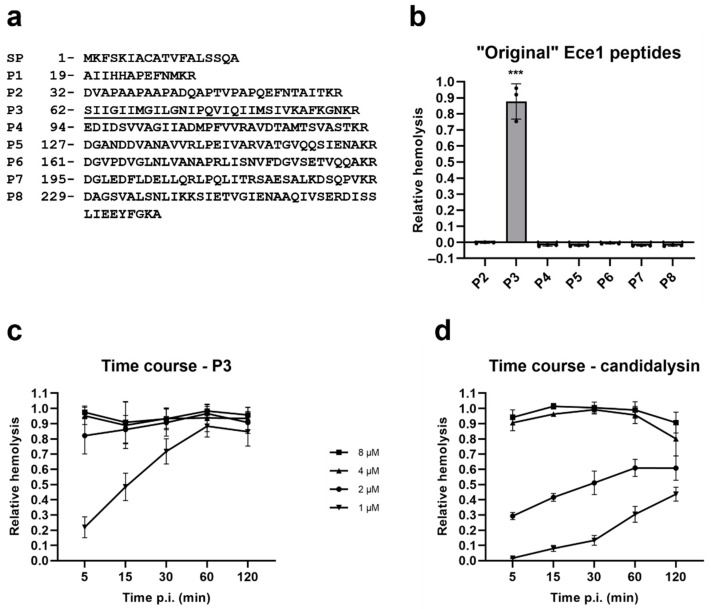
Candidalysin and its precursor P3 are hemolytic toxins. (**a**) Amino acid sequences of Ece1 peptides. Candidalysin’s sequence is underlined. Numbers represent the position of the first amino acid on the Ece1 sequence. (**b**–**d**) Purified RBCs were incubated with (**b**) 9 µM of the “original” Ece1 peptides (defined by the Kex2 processing at KR sites), (**c**) increasing concentrations of P3 in a time course experiment, and (**d**) increasing concentrations of candidalysin in a time course experiment. RBCs and peptides were incubated at 37 °C for 1 h (**b**), or for the indicated time (**c**,**d**). Hemolysis was quantified by measuring the absorbance of sample’s supernatant at 414 nm, and plotted relative to the full lysis control sample (RBCs incubated with pure water), following subtraction of the vehicle control. Each data point in (**b**) represents a different donor (average of 2 technical replicates). Error bars show the standard deviation. In (**c**,**d**) data is presented as the mean of 4 different donors ± standard deviation. For statistical analysis (only in (**b**)), an arbitrary value of 0.01 was assigned to any value that was below this threshold. Student’s paired *t*-tests were then performed on log transformed data vs. vehicle control only. ***, *p* < 0.005.

**Figure 3 toxins-14-00874-f003:**
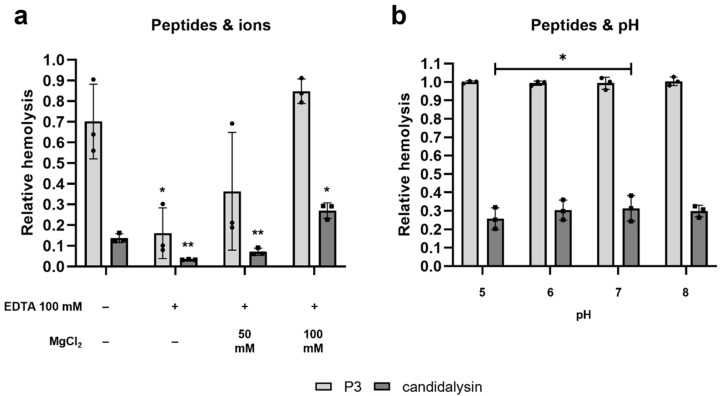
Candidalysin and P3-induced hemolysis is ion-dependent and pH-independent. Purified RBCs were incubated with candidalysin or P3 (9 µM, *Proteogenix*) for 1 h at 37 °C. (**a**) ± EDTA ± MgCl_2_ at the indicated concentrations. (**b**) In buffered Hank’s balanced salt solution (HBSS) media at different pHs. Hemolysis was quantified by measuring the absorbance of sample’s supernatant at 414 nm, and plotted relative to the full lysis control sample (RBCs incubated with pure water), following subtraction of the vehicle control. Each data point on the graph represents a different donor (average of 2 technical replicates). Error bars show the standard deviation. For statistical analysis, data were log-transformed and in (**a**) a student’s paired *t*-tests, vs. the relative peptide-only sample, was performed. In (**b**), a repeated measures one-way ANOVA analysis was performed, with a post hoc Tukey’s multiple comparison test across samples treated with the same peptide *, *p* < 0.05; **, *p* < 0.01.

**Figure 4 toxins-14-00874-f004:**
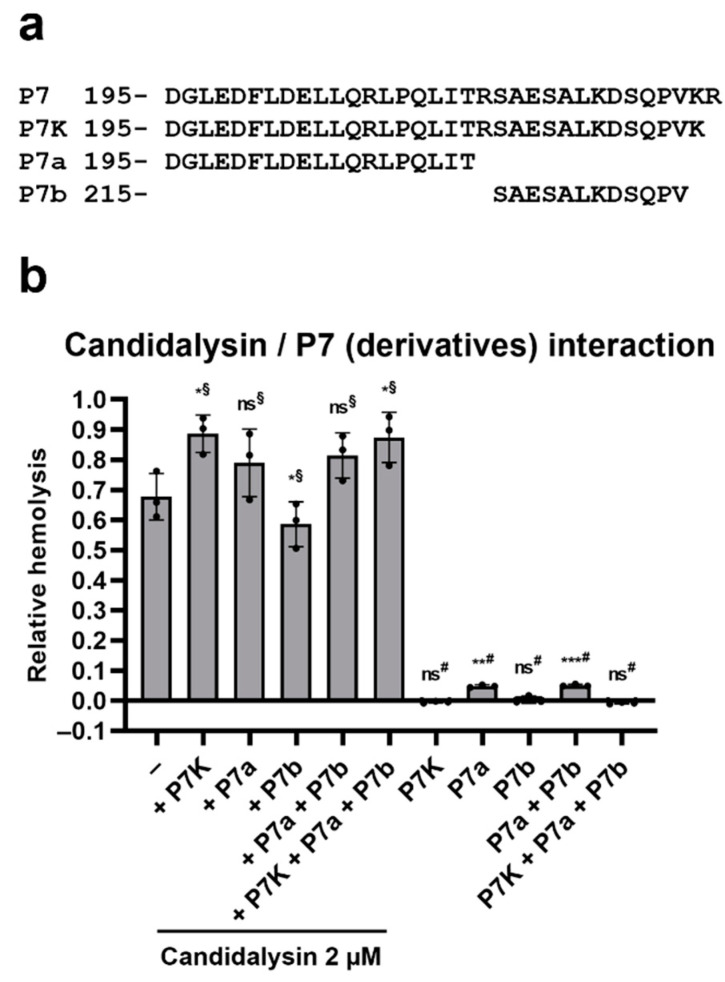
P7 derivatives modulate the hemolytic potential of candidalysin. (**a**) Amino acid sequences of P7-derived peptides. Numbers represent the position of the first amino acid respective to the N-terminus of Ece1. (**b**) Purified RBCs were incubated with candidalysin and P7K, P7a and P7b (each peptide at 2 µM, Peptide Protein Research Ltd.) for 1 h at 37 °C. Hemolysis was quantified by measuring the absorbance of sample’s supernatant at 414 nm, and plotted relative to the full lysis control sample (RBCs incubated with pure water), following subtraction of the vehicle control. Each data point on the graph represents a different donor (average of 2 technical replicates). Error bars show the standard deviation. For statistical analysis, an arbitrary value of 0.01 was assigned to any value that was below this threshold. Student’s paired *t*-tests were then performed on log-transformed data. ^§^, *t*-test vs. candidalysin only; ^#^, *t*-test vs. vehicle only. *, *p* < 0.05; **, *p* < 0.01; ***, *p* < 0.005; ns, not significant.

**Figure 5 toxins-14-00874-f005:**
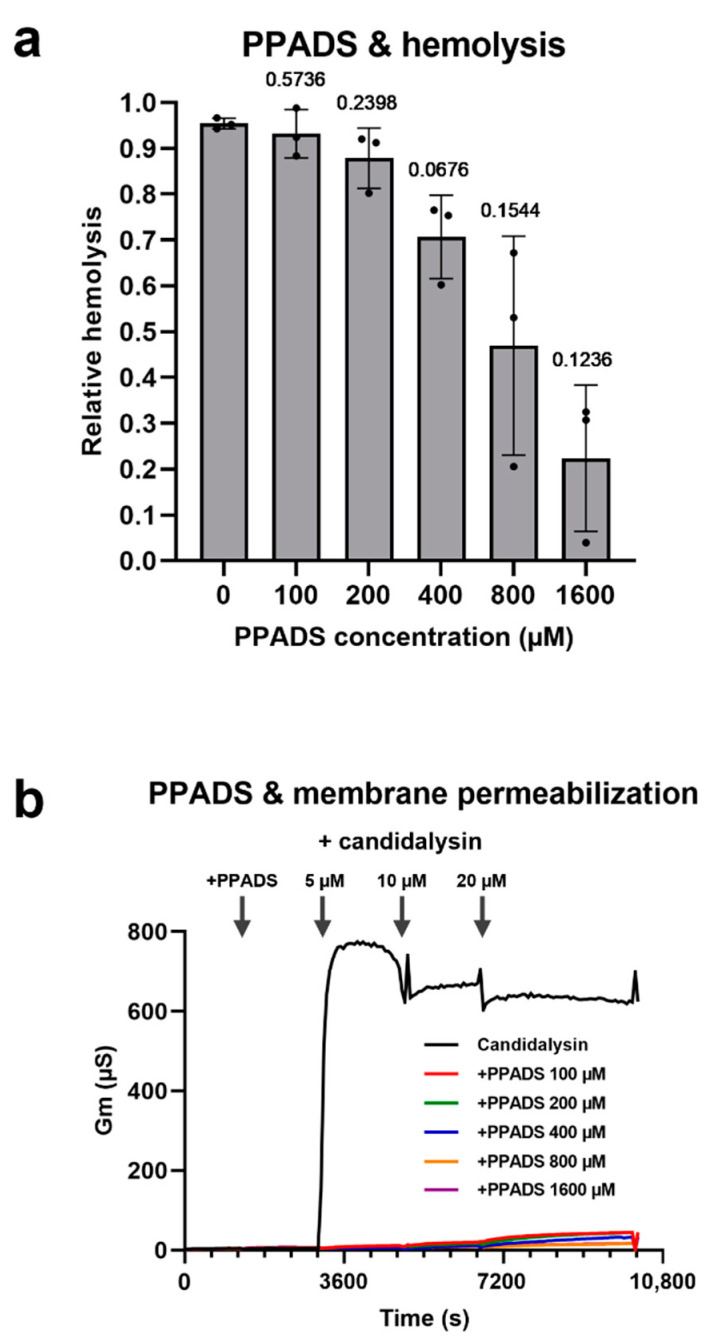
The purinergic receptor antagonist pyridoxal-phosphate-6-azophenyl-2′,4′-disulfonic acid (PPADS) protects from candidalysin-induced hemolysis via a purinergic receptor-independent mechanism. (**a**) Purified RBCs were incubated with candidalysin (16 µM) and increasing concentrations of PPADS, for 1 h at 37 °C. Hemolysis was quantified by measuring the absorbance of sample’s supernatant at 414 nm, and plotted relative to the full lysis control sample (RBCs incubated with pure water and the same amount of PPADS), following subtraction of the relative vehicle control (containing the same amount of PPADS). Each data point on the graph represents a different donor (average of 2 technical replicates). Error bars show the standard deviation. For statistical analysis, student’s paired *t*-tests, vs. the candidalysin only sample, were then performed on log-transformed data. *p* values are depicted on top of each bar. (**b**) Synthetic membranes were incubated with the same increasing concentrations of PPADS as in (**a**) followed by addition of increasing concentrations of candidalysin, at 37 °C. Permeabilization was measured as kinetic changes in conductance of tethered lipid membranes.

**Figure 6 toxins-14-00874-f006:**
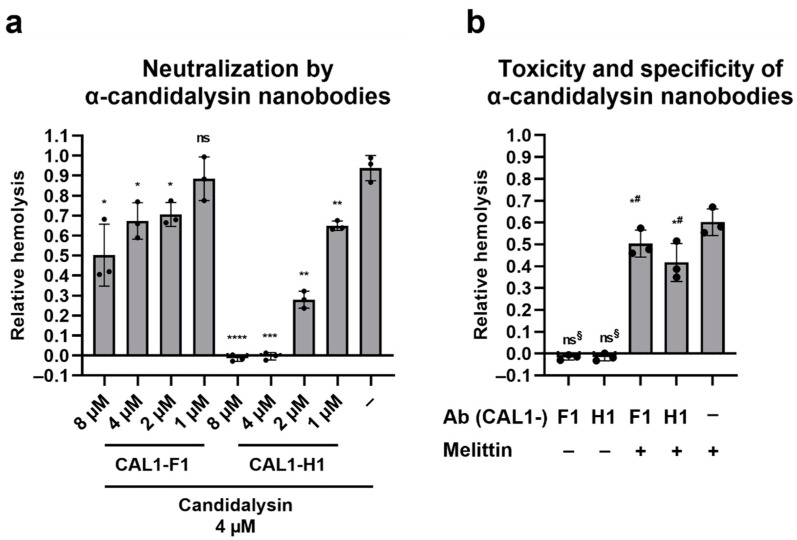
Llama-derived anti-candidalysin nanobodies neutralize the hemolytic potential of candidalysin. (**a**) Candidalysin (4 µM) was pre-incubated with different concentrations of the two nanobody clones (CAL1-F1/CAL1-H1) for 1 h. (**b**) Melittin (4 µM) was preincubated with 8 µM of the two nanobody clones. The mixture was then added to the purified RBCs, and hemolysis was quantified after 1 h incubation at 37 °C, by measuring the absorbance of sample’s supernatant at 414 nm, and plotted relative to the full lysis control sample (RBCs incubated with pure water), following subtraction of the vehicle control. Each data point on the graph represents a different donor (average of 2 technical replicates). Error bars show the standard deviation. For statistical analysis, an arbitrary value of 0.01 was assigned to any value that was below this threshold. Student’s paired *t*-tests were then performed on log transformed data. (**a**) *t*-test vs. candidalysin only; (**b**) *t*-test vs. vehicle control only (^§^) or vs. melittin only (^#^). *, *p* < 0.05; **, *p* < 0.01; ***, *p* < 0.005; ****, *p* < 0.001; ns, not significant.

**Table 1 toxins-14-00874-t001:** Strains used in this study.

Strain Name	Description	Internal Name	Relevant Genotype	References
Wt (isogenic)	BWP17/CIp30 Isogenic wild type	M1477	*ura3*::λimm434/*ura3*::λimm434 *iro1*::λimm434/*iro1*::λimm434 *his1*::*hisG*/*his1*::*hisG* *arg4*::*hisG*/*arg4*::*hisG* *RPS1*/*rps1*::(*URA3*-*HIS1*-*ARG4*)	[21]
*ece1*Δ/Δ	*ece1* double knock-out mutant	M2057	*ece1*::*ARG4*/*ece1*::*HIS1* *RPS1*/*rps1*::*URA3*	[5]
*ece1*Δ/Δ + *ECE1*	*ECE1* revertant	M2059	*ece1*::*ARG4*/*ece1*::*HIS1* *RPS1*/*rps1*::(*URA3*-*ECE1*)	[5]
ΔP3	Mutant lacking only the P3 sequence (all other *Ece1* peptides expressed)	M2174	*ece1*::*ARG4*/*ece1*::*HIS1* *RPS1*/*rps1*::(*URA3*-*ECE1*^Δ184–279^)	[5]

## Data Availability

Data are contained within the article.

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
