# Peer review of "Candidalysin Is the Hemolytic Factor of Candida albicans"

_toxins, 2022, doi:10.3390/toxins14120874_

Round 1
Reviewer 1 Report
The manuscript addresses an interesting subject that fits well in this journal. Overall, the study is well-designed, and the results are clear and support the main conclusions.
I have the following observations for manuscript improvement:
The manuscript should include an ethical statement, indicating the approval code and committee that provide it, and state whether signed informed consent was obtained before blood withdrawal.
The manuscript should include a section describing the statistical analysis applied to the data. In figure 1, the authors indicated that a T-student analysis was used for data analysis. I do not think this is the most appropriate analysis. The authors should use non-parametric tools for results analysis.
Panel A in figure 1 adds not much to the results. I suggest moving it to supplementary material.
In Figure 1, why did the revertant strain show a partial restoration of the phenotype?
Reviewer 2 Report
Here, the authors describe the haemolytic activity fro candidalsyin for the first time. This is an important contribution to the field. The manuscript is well-written and clear. Only minor experimental amendments remain that should be feasible to be addressed.
Major comments:
Fig 2c and d: In c) relative lysis at 6µM candidalysin below 0.1 in d) it is at 2 µM above 0.4, why is that? Differences in haemolytic activity of candidalysin and P3 should be explored. The authors allude and speculate that the discrepancy might stem from different suppliers of the toxin. However, if the sequence is correct and identical for both peptide preparations, there should not be a difference. They could only result from impurities which should not be present to prevent unwanted bystander effects. Hence, the experiments should be repeated with the same batches of candidalysin and P3. This is a fairly easy experiment for the group to conduct and is very important since the differences in haemolytic activity of P3 and candidalysin are a central part in the study.
Fig 4: It remains elusive what the conclusions are from the experiments with P7-dervied peptides. Since the effect is minor the authors argue that the main hemolytic activity stems from candidalysin alone. So why show the data then? If the authors insist to include the data, they should better motivate the inclusion.
Fig 5b: The result of the experiment with artificial membrane is not conclusive. Any concentration of the inhibitor leads to a complete block of the change in membrane conductance. However, wt these concentrations there is no inhibition of haemolytic activity observed. This points toward a flaw in the membrane experiment. Could it be that the inhibitor has an effect on conductivity? It would be advisable to chose another read out to measure the effect of the inhibitor on artificial membranes.
Fig 6. The figure legend is insufficient. It remains unclear which concentrations were used in 6b) for nano-bodies and toxins.
Minor comments
None
Round 2
Reviewer 2 Report
All issues raised were addressed adequately.